# Herpes Simplex Virus, Human Papillomavirus, and Cervical Cancer: Overview, Relationship, and Treatment Implications

**DOI:** 10.3390/cancers15143692

**Published:** 2023-07-20

**Authors:** Daniel G. Sausen, Oren Shechter, Elisa S. Gallo, Harel Dahari, Ronen Borenstein

**Affiliations:** 1Department of Microbiology and Molecular Cell Biology, Eastern Virginia Medical School, Norfolk, VA 23501, USA; sausendg@evms.edu (D.G.S.); shechto@evms.edu (O.S.); 2Division of Dermatology, Tel Aviv Sourasky Medical Center, Tel Aviv 64239, Israel; 3The Program for Experimental and Theoretical Modeling, Division of Hepatology, Department of Medicine, Stritch School of Medicine, Loyola University Chicago, Maywood, IL 60153, USA; hdahari@luc.edu

**Keywords:** HSV, HPV, oncogenic virus, cervical cancer, cervical cancer treatment

## Abstract

**Simple Summary:**

Many different viruses are known to cause cancer. It is well known that human papillomavirus is a causative agent of cervical carcinoma. What is more controversial in the pathogenesis of cervical cancer is the role of herpes simplex virus. If herpes simplex does contribute to cervical cancer, then it could become a therapeutic target against a disease that is responsible for hundreds of thousands of deaths per year. After a brief introduction to both viruses, this review assesses the evidence for and against the involvement of herpes simplex in the development of cervical cancer. Furthermore, it discusses treatment options for this disease, including preventative measures and recent therapeutic advances.

**Abstract:**

There is a significant body of research examining the role of human papillomavirus (HPV) in the pathogenesis of cervical cancer, with a particular emphasis on the oncogenic proteins E5, E6, and E7. What is less well explored, however, is the relationship between cervical cancer and herpes simplex virus (HSV). To date, studies examining the role of HSV in cervical cancer pathogenesis have yielded mixed results. While several experiments have determined that HPV/HSV-2 coinfection results in a higher risk of developing cervical cancer, others have questioned the validity of this association. However, clarifying the potential role of HSV in the pathogenesis of cervical cancer may have significant implications for both the prevention and treatment of this disease. Should this relationship be clarified, treating and preventing HSV could open another avenue with which to prevent cervical cancer. The importance of this is highlighted by the fact that, despite the creation of an effective vaccine against HPV, cervical cancer still impacts 604,000 women and is responsible for 342,000 deaths annually. This review provides an overview of HSV and HPV infections and then delves into the possible links between HPV, HSV, and cervical cancer. It concludes with a summary of preventive measures against and recent treatment advances in cervical cancer.

## 1. Introduction

Cervical cancer was estimated to afflict 604,000 women in 2020. In fact, it ranks as the fourth most common malignancy in women and represents a significant cause of mortality, with 342,000 deaths in 2020 [1]. Some symptoms of cervical cancer include dysuria, post-coital bleeding, irregular menstruation/amenorrhea, painful intercourse, and vaginal discharge [2]. Notably, higher-income countries have lower rates of cervical cancer than lower/middle-income ones because of more robust vaccine and screening programs [3].

Cervical cancer is subdivided into stages I–IV [4]. According to the International Federation of Gynecology and Obstetrics (FIGO), stage I carcinoma occurs when there is no spread beyond the cervix. Stage II disease occurs when there is spread beyond the uterus, but it does not involve the lower third of the vagina or the pelvic wall. This contrasts with stage III cancer, which can involve the lower third of the vagina or pelvic wall. Additional pathologic features consistent with stage III disease include related hydronephrosis, renal failure, or pelvic and/or paraaortic nodal involvement. Stage IV disease pertains to spread beyond the true pelvis or when there is biopsy-proven involvement of the bladder/rectal mucosa [4]. The stages are summarized below in Figure 1.

Nearly all cases of cervical cancer are caused by human papillomavirus (HPV) [5]. The vast majority of cervical cancers are classified as squamous cell carcinoma, although other histologic types include adenocarcinoma, adenosquamous, and less frequently clear cell, neuroendocrine, small cell, and serous papillary carcinoma [6]. HPV is an oncogenic virus that has been associated with multiple other cancers, including head, neck, and anogenital malignancies [7]. Mechanistically, the HPV proteins E6 and E7 inactivate p53 and Rb, respectively, which inhibits cell cycle regulation and leads to cell proliferation [7]. In addition to its role in carcinogenesis, HPV causes both cutaneous and anogenital warts [8].

Studies have questioned the potential role of other viruses in cervical cancer pathogenesis. For example, a recent association has been reported between hepatitis B infection (i.e., positive for hepatitis B surface antigen, HBsAg) and cervical cancer in patients younger than 50 years of age [9]. Compared with the benign controls, younger patients with both hepatitis B and HPV infections had an increased risk of cervical cancer (adjusted odds ratio of 67.1; 95% CI 23.4–192.7) [9]. In contrast, Feng et al. [10] did not find such an association, but surprisingly suggest that a history of hepatitis B infection (defined as patients with HBV testing profiles of anti-hepatitis B core antigen, HbcAb-positive, and HBsAg-negative or HbcAb-positive and HBsAb-positive) was associated with better outcomes in cervical cancer [10]. In agreement with that, hepatitis B vaccination was reported as a positive predictor of cervical cancer screening [11], suggesting that acquired immunity for hepatitis B may contribute to a better prognosis secondary to earlier detection. Questions have also been raised about the potential contributions of sexually transmitted infections (STI), such as chlamydia trachomatis [12] and other infections that lead to cervicitis/cervical infection [13]. One such virus that may contribute to cervical cancer pathogenesis is herpes simplex virus (HSV), although evidence for a causal relationship is mixed [14,15].

HPV is a non-enveloped, icosahedral virus with circular, double-stranded DNA and a surrounding capsid comprised of L1 and L2 proteins [16]. It has a marked predilection to infect squamous cells in the epidermal basal layer [16]. The virus is predominantly spread by contact through sexual transmission; however, other modalities have been suggested [17], including vertical transmission [18], autoinoculation [19], and surgical smoke [20].

HSV-1 and HSV-2 are enveloped viruses with an icosahedral capsid containing double-stranded DNA [21]. They are two of the eight identified human herpesviruses comprising the Herpesviridae family [22]. Like HPV, HSV is an extremely common STI responsible for approximately 500 million genital infections and billions more oral infections. In 2016, it was estimated that more than 3.75 billion individuals were infected by HSV-1 and nearly 500 million were infected with HSV-2 [23]. While HSV-2 is more commonly implicated, HSV-1 is the causative agent in a relatively small but growing number of genital infections [24]. HSV-1 is characterized by vesicular eruptions that mainly affect mucosa in the orolabial and genital regions [25]. HSV-2 infection is also characterized by the formation of vesicular lesions as well as non-specific symptoms, including itching, irritation, and excoriation [24]. Other HSV disease manifestations include herpes stromal keratitis, encephalitis, and meningitis [26].

Histopathological findings of herpes infection include acantholysis, ballooning degeneration, intranuclear inclusions, multinucleated cells, necrosis and the formation of vesicles and/or ulcers [27]. There are no approved vaccines available for HSV prevention [28].

This paper will summarize the evidence for the role of HSV in conjunction with HPV in the pathogenesis of cervical cancer. It will begin by providing a brief overview of HSV and HPV infections, after which it will examine the links between these viruses and malignancies of the cervix. It will conclude with a discussion of therapeutic advances in the treatment of these malignancies.

## 2. Overview of HPV Infection and Oncogenicity

HPV is the most common sexually transmitted infection (STI) in the United States [29]. More than 200 strains of HPV have been discovered to date, of which a subset are considered high risk [30]. Oncogenic strains include the following: 16, 18, 31, 33, 35, 39, 45, 51, 52, 56, 58, 59 and 66 [31]. HPV-53 is also considered potentially high risk, but it has low pathogenicity and slow rates of progression [32]. Of these high-risk varieties, HPV-16 and HPV-18 are most strongly associated with cancer occurrence [33,34], and HPV-16 is a relatively common strain [35]. In fact, one study of 1163 Koreans found that HPV-16 was the most commonly seen strain, affecting 12.3% of the sample. HPV-58, another high-risk strain, was the second most common [35]. A separate study examining the frequency of various genotypes in Western China demonstrated that HPV-16 was the second most common genotype behind HPV-52, while HPV-58 was the third most common. All three genotypes were associated with cytologic abnormalities of the cervix [36].

Infection begins following trauma to the epithelial barrier, which provides a means for the virus to access the dividing keratinocytes of the basal epithelial layer [37,38]. The majority of new HPV infections are suppressed within 1–2 years of infection. While this may represent the resolution of the infection, it is also possible that HPV enters a latent state [39]. It is during ongoing infection that HPV employs several strategies to facilitate immune evasion, such as reducing the amount of antigen being produced during the early phase of infection, suppressing TLR9 expression in keratinocytes, and downregulating NF-κB [40]. Other strategies employed by HPV include recruiting regulatory T cells, preventing IFN synthesis, and inhibiting apoptosis via p53 inactivation [41]. Ongoing detectability is typically associated with an elevated risk of progression into precancer (either as high-grade squamous intraepithelial lesion [HSIL] or cervical intraepithelial neoplasia [CIN] grades 2 or 3) [42,43]. Progression from HPV infection to cancer often can take more than a decade [39,44], with one study estimating that it takes 23.5 years for CIN 2 or 3 to progress to cervical cancer [44].

The HPV genome includes a long control region (LCR), an early gene region encoding E1, E2, E4, E5, E6, and E7, and a late gene region encoding L1 and L2. Figure 2 below depicts the genome structure of HPV. The capsid proteins L1 and L2 play key roles in viral entry [45], with the major capsid protein L1 binding to host cell heparan sulfate proteoglycans [46]. L2 plays a key role in virion trafficking [47]. While L1 is not as heavily implicated, it has been suggested that L1 plays a role in stabilizing the HPV genome during this process [48]. Following binding, the host cell surface protein Cyclophilin B promotes a conformational change that exposes the L2 N terminus [49].

Tetraspanin CD151 and CD151-associated integrins such as α3β1 and α6β4 are also implicated in HPV entry [50]. Tetraspanins are transmembrane glycoproteins that can form tetraspanin-enriched microdomains, which allow for interactions between multiple subtypes of proteins. They play roles in multiple cellular processes, including migration, signaling, and adhesion [45]. Notably, numerous viruses take advantage of these tetraspanins for host cell entry [45], including human cytomegalovirus [51] and hepatitis C virus [52]. HPV is thought to use the cytoskeletal adaptor protein obscurin-like protein 1, which associates with tetraspanins, to link the virus contact point to intracellular endocytic actin [53]. Other host factors that play a role in HPV entry include syndecan-1 [54] and annexin A2 heterotetramer [55]. In addition, ADAM17 stimulates the formation of a viral entry platform comprised of CD151, epidermal growth factor receptor (EGFR), and HPV-16 via extracellular signal-regulated kinase (ERK) signaling [56].

Once inside the cell, the minor capsid protein L2 facilitates the transfer of HPV DNA to the host nucleus [57]. Recent research has demonstrated that entry of the L2/viral DNA complex into the nucleus is dependent on the progression of the cell cycle. After the viral DNA enters the nucleus, it associates with condensed chromosomes, thus allowing it to spread to cell progeny [58]. Individual viral entry and life cycle studies cited above use HPV-16 as a model.

Viral replication occurs in the nucleus in three distinct phases. In the first phase, the viral genome replicates quickly to increase episome copy numbers. This allows the virus to establish infection in basal cells. Only the early promoter is active at this time. The second phase, or stable maintenance phase, is synchronized with chromosomal replication and involves maintaining the number of viral genomes [59]. This phase occurs in undifferentiated cells that are actively dividing [60]. Differentiation is required to enter the productive phase of viral replication. This third phase, called vegetative amplification, involves a rapid increase in viral genome numbers and the activation of the host cell DNA damage response, which is harnessed by the virus to enhance replication [59]. Vegetative amplification only occurs in differentiated cells and not in cells of the basal layer [61]. Indeed, working with HPV-31, Spink et al. showed that differentiation is required for late promoter induction, which supports the idea that vegetative amplification can only occur in differentiated cells [62]. Indeed, differentiation is required for the expression of late genes, and the immunogenic proteins L1 and L2 are only expressed in differentiated cells [60]. Given that HPV does not have a lytic phase, virions are only released once infected cells attain the surface epithelium [63]. Figure 3 below summarizes the HPV life cycle. HPV undergoes theta replication. Using HPV-11, Orav et al. demonstrated that this requires interactions between the E1 protein of HPV-11 and a member of the Fanconi anemia DNA repair pathway, the cellular UAF1-USP1 complex [64]. There is also a second, less well-characterized mechanism of replication that proceeds unidirectionally that has been observed in HPV-18 [65].

The E1 and E2 proteins are heavily involved in the early phases of viral replication [66,67]. E1 is a DNA helicase/ATPase [68] that is best known for its role in the initial amplification of HPV DNA [69], although it may play a role in select situations during other phases of HPV infection [69]. Emerging evidence has supported a more varied role for E1. The E1 protein from HPV-16 and HPV-18 were shown to downregulate the host immune response, including the key antiviral mediators IFNβ1 and IFNλ1 [70].

The viral protein E2 also plays a key role in replication [66,71]. It acts as a transcription factor that recruits E1 to the viral origin of replication [72]. E2 is a key transcriptional regulator, and recent evidence from experiments using HPV-31 shows that it interacts with the short form of bromodomain-containing protein 4 (BRD4S) to prevent the expression of late genes [73]. Following transfection with E2, the number of population doublings in human foreskin keratinocytes increased from 8 to 23 before they entered senescence. This increase in doublings as a result of transfection indicates that high-risk E2 may play a role in cell immortalization [74]. Notably, E1, E2, and the origin of replication are the only viral components required for early replication of the HPV genome. The rest of the replication machinery is provided by the host cell [75].

E4 expression is synchronized with and plays a role in the onset of vegetative replication. It also modifies the host cells to promote viral reproduction [76,77]. The pE8 promoter controls the expression of the protein E8^E2 [31], which is an important negative regulator of HPV gene expression and replication [78,79]. Mechanistically, it was shown that the cellular NCoR/SMRT repressor complex, including the proteins HDAC3, GPS2, NCoR, SMRT, TBL1, and TBLR1, is essential in carrying out this inhibitory function [80].

E5, E6, and E7 are the primary oncogenic proteins in HPV [81,82]. They are responsible for the creation of infectious particles [83]. They also alter the cell cycle, which promotes hyperproliferation and subsequent carcinogenesis [82,83]. These proteins will be discussed further shortly.

## 3. Overview of HSV Infection

HSV-1 and HSV-2 are members of the Herpesviridae family, which also includes varicella zoster virus (VZV) [84]. HSV is widespread globally, with HSV-1 estimated to infect over 3.752 billion persons (approximately 67% of the population) and HSV-2 estimated to infect over 491.5 million persons (approximately 13% of the population) worldwide as of 2016 [23]. HSV-1 is highly contagious, with infection commonly occurring during childhood [85]. Typically, HSV-1 is transmitted orally via contact with cold sore lesions that shed viral particles or through secretions that may occur during asymptomatic shedding [86]. HSV-2 is typically transmitted through viral shedding that occurs primarily via genital contact and is considered to be a leading cause of sexually transmitted infections [87]. While symptoms of HSV-1 and HSV-2 infection are similar, they differ in aspects such as the site of CNS infection and their reactivation rate [88]. Once HSV infects a person, the virus persists throughout the lifespan of the infected individual and can manifest as oral and genital ulcers, herpetic keratitis, and encephalitis [89]. Virus replication occurs in epithelial cells, with latency occurring in sensory neurons [90]. The latent site of infection for herpes labialis is the trigeminal ganglia, while the latent site for herpes genitalis is the sacral ganglia [91]. Latent HSV undergoes very limited gene expression and does not produce infectious virus particles [26]. Indeed, the only product demonstrating significant expression during latency is HSV latency-associated transcript (LAT), a long noncoding RNA that represses gene expression and acts as a key regulator of viral latency [92,93].

HSV has a linear, double-stranded DNA genome surrounded by an icosahedral capsid. Tegument proteins surround the capsid, and a lipid bilayer with proteins and glycoproteins forms the outer layer of the virion [94]. The virion structure is depicted below in Figure 4. Most experiments examining the viral cycle of HSV have focused on HSV-1; however, HSV-2 infection is thought to progress in a similar fashion [87].

Attachment and viral entry are primarily mediated by the four glycoproteins gB, gD, and gH/gL [95]. HSV can also enter through an endocytic path in which the low pH of the endosome stimulates conformational changes in HSV fusion proteins and leads to fusion with the vesicular membrane [95]. Once the virus successfully enters the cell, it is transported toward the nucleus by subverting the dynein motor complex and host microtubules. Key viral mediators of this subversion include UL37, which interacts with the dynein intermediate chain, and ICP5 (VP5), which interacts with dynactin [96]. Entry into the nucleus requires importin β and Ran GTPase to bind to the nuclear pore [97]. The tegument protein VP1-2 then undergoes proteolytic cleavage, which allows for conformational changes in the capsid that facilitate entry [98].

Once inside the nucleus, DNA replication can occur. While HSV-1 encodes many proteins required for DNA replication, such as an origin-binding protein, DNA polymerase, helicase/primase complex (UL9, UL30, and UL5/UL8/UL52, respectively), and others [99], it also co-opts a number of host cell proteins [99,100]. While the protein components of the capsid are generated by cytosolic ribosomes, capsid assembly and DNA packaging occurs in the nucleus [101]. Completed capsids exit the nucleus by budding through the inner nuclear membrane and subsequent fusion with the outer nuclear membrane, potentially with initial tegument proteins attached [102]. Once tegument assembly is completed in the cytoplasm, the virus undergoes secondary envelopment in which the virus is surrounded by a lipid bilayer and then exits the cell [102].

A key step in establishing infection is the degradation of host mRNA. This degradation of host mRNA results in the inhibition of host cell protein synthesis and is mediated by UL41, also called virion host shutoff protein [103,104]. Mechanistically, UL41 acts as an endonuclease with targets similar to ribonuclease A [104]. The decreased capacity for host cell protein synthesis allows for enhanced efficacy of viral mRNA translation [105].

## 4. Relationship between HPV and Cervical Cancer

Dr. Harald zur Hausen first espoused the theory that HPV was linked to cervical cancer in 1976 [106]. Since then, the link between HPV and cervical malignancy has been well established, with it being estimated that HPV is responsible for approximately 99.7% of cases of cervical cancer [5]. While more than 200 types of HPV exist [107], they are broadly categorized into two groups, low-risk HPV types that can cause genital warts and high-risk HPV types that are oncogenic in nature [108]. There are 14 high-risk strains of HPV (HPV 16, 18, 31, 33, 35, 39, 45, 51, 52, 53, 56, 58, 59, and 68) [109]. Of these 14 high-risk strains, two strains, in particular, HPV-16 and HPV-18, account for approximately 70% of cervical cancer cases [110].

Several oncoproteins play a role in the pathogenesis of HPV-mediated cervical cancer, the most important of which are the E5/6/7 oncoproteins [111]. The structure of the HPV16 genome is 7.9 kb long and is divided into an early gene-coding region (E), a late coding region (L) and the long control region (LCR) [81]. In the early gene-coding region, there are six open reading frames, E1/E2/E4/E5/E6/E7 [81]. While E1/E2 play roles in controlling viral genome replication as well as early protein transcription, E5/E6/E7 are considered to be the main facilitators of oncogenesis [81]. 

E5 promotes oncogenesis through various mechanisms, such as disrupting the acidification of endosomes, thus enhancing EGF receptor recycling [112]. E5 also plays a role in promoting the upregulation of Met, a receptor tyrosine kinase that is implicated in facilitating tumor cell invasion [113]. In addition, E5 cooperates with E6 and E7 to enhance numerous pro-carcinogenic features. HaCaT cells transduced with E5/E6/E7 have more viable cells and proliferate faster than control cells [114]. Moreover, proliferation rates are faster in cells transfected with all three oncoproteins than those transfected with E5 or E6/E7 [114]. Cells transfected with E5/E6/E7 are also more invasive than control cells and those transfected with E5 or E6/E7 [114]. Transduction with the three oncoproteins alters the cellular redox state too; these cells have higher rates of peroxiredoxin when compared to control and E5 transduced cells. They also have a greater expression of the antioxidant GSH than all other cells [114]. Notably, high-risk and low-risk E5 proteins have different roles in HPV infection, and the difference is mediated by just two amino acids [115].

E6 promotes the degradation of p53 [116,117]. E6 also promotes the upregulation of IL-6, which in turn activates the JAK-STAT pathway and may stimulate a pro-inflammatory, pro-proliferative microenvironment [118]. Experiments with HPV-18 show that E6-mediated induction of STAT3 is essential in driving the cell cycle. Loss of STAT3 expression inhibits HPV gene expression and episome maintenance [119].

Recent research has analyzed other molecular targets of the E6 protein. Fan et al. demonstrated that HPV-16 E6 was capable of binding to the APOBEC3B promoter to upregulate its expression. They showed that this protein was highly expressed in HPV-16/-18-associated cervical cancer and was associated with the development of metastatic disease [120]. Importantly, APOBEC3B expression was found to result in hypomethylation of the promoter for *CCND1*, which codes for cyclin D1 [120]. Cyclin D1 is a key regulator of the cell cycle and controls the G1/S transition [121].

The c-Jun N-terminal kinase (JNK) pathway is another target modulated by E6 [122]. JNK1/2 phosphorylation increases in cervical cancer tissue when compared to control tissue. E6 stimulates JNK1/2 phosphorylation through the E6 PDZ-binding motif [122]. JNK inhibition results in a concomitant decrease in c-Jun phosphorylation and expression, as well as a decrease in cell growth. This inhibition is not noted in the HPV-negative cell line C33A [122]. The JNK/c-Jun pathway promotes the expression of transcription factors required for epithelial/mesenchymal transition (EMT), such as Slug. It also promotes the expression of the mesenchymal marker vimentin. Matrix metalloproteinase 9, a key pro-invasive enzyme, is downregulated following JNK inhibition. This signaling path is needed for the constitutive expression of E6 and E7 [122]. Mechanistically, the activation of the JNK pathway leads to increased EGFR signaling, which in turn leads to cell survival, proliferation, and EMT [122].

Low levels of the microRNA (miR) hsa-miR-504 are associated with both the development of cervical cancer and a poor prognosis [123]. This represents another target of the E6 protein [124]. E6 overexpression in the cervical cancer line SiHa results in significantly lower levels of miR-504. Overexpression of E6 also augments the proliferative and invasive abilities of SiHa cells and inhibits apoptosis, changes reversed with miR-504 overexpression [124].

Recent research has begun to explore the role of long non-coding RNAs (lncRNA) in carcinogenesis [125,126]. One such lncRNA implicated in the pathogenesis of cervical cancer is lnc_000231, which is upregulated in cervical cancer [127]. E6 upregulates the expression of lnc_000231. The mechanism through which this upregulation occurs involves promoter H3K4me3 modification. Specifically, E6 destabilizes the histone demethylase KDM5C. The increased lnc_000231 expression results in lower levels of miR-497-5p, which in turn results in higher expression levels of cyclin E1, which is involved in the G1/S transition [128]. Notably, lnc_000231 knockdown stunts cell growth, cell colony formation and cell cycle progression in vitro and inhibits tumor formation in vivo [128].

E7 plays a role in promoting IL-6 expression, although its effect is less than E6 [118]. The E7 oncoprotein targets pRB and facilitates its degradation, causing activation of E2F transcription factors and subsequent downstream genes promoting cell proliferation [129]. Moreover, E7 can impair p53 function even in the absence of E6 [130].

Like with E6, recent research analyzed novel molecular targets of E7. Phosphorylated AKT and phosphorylated SRC both have higher levels of expression in cervical cancer, including higher levels in invasive samples than in precancers [131]. Knockdown of HPV-16 E7 protein results in diminished expression of phosphorylated AKT (p-AKT) and phosphorylated Src (p-SRC), while E7 expression results in increased levels of these two proteins. Notably, E7 is continually expressed to maintain elevated levels of p-AKT and p-SRC. Indeed, HPV-16 E7 leads to the expression of p-AKT and p-SRC, which then stimulates the initiation and progression of cervical cancer [131].

In addition, high-risk E7 was shown to stimulate cervical cancer cell proliferation and migration. It increases levels of topoisomerase II α (TOP2A), BIRC5, and E2F1 [132]. Inhibition of E2F1 results in decreased levels of TOP2A and BIRC5 and subsequent inhibition of migration and proliferation of cervical cancer cells. E7 results in the upregulation of E2F1, which subsequently enhances BIRC5 and TOP2A expression [132]. TOP2A expression is positively correlated with cervical cancer [133] and is overexpressed in many human malignancies [134]. BIRC5, also known as survivin, is an anti-apoptotic protein that is common in cancer [135].

E7 targets the promoter of the lncRNA metastasis-associated lung adenocarcinoma transcript 1 (MALAT1) [136], a lncRNA that has been associated with cervical cancer progression [137,138]. Overexpression of HPV-16 E7 increases MALAT1 when transfected into HPV-negative HEK-293T and C33A cells, and E7 knockdown in the human cervical carcinoma lines CaSki and SiHa results in decreased MALAT1 expression [136]. Subsequent experiments demonstrate that this effect is mediated through the MALAT1 promoter [136]. Notably, small interfering RNA (siRNA) targeting MALAT1 reverses the enhancements in cell proliferation, invasion, and migration mediated by E7 [136]. E7 also enhances the expression of lnc-EBIC [139]. Overexpression of this lnc in HPV-negative C33A cells enhances proliferation, migration, invasion, and survival [139]. In addition, lnc-EBIC results in enhanced expression of the oncogenic Kelch domain-containing 7B (KLHDC7B), a finding corroborated by the fact that KLHDC7B knockdown strongly inhibits lnc-EBIC-mediated tumorigenesis in C33A cells [139]. 

## 5. Relationship between HSV and Malignancy

The carcinogenic properties of several human herpesviruses have long been known [140,141,142,143,144]. Epstein–Barr virus (EBV) was the first virus discovered to have oncogenic properties in humans [145]. Thus, it is perhaps unsurprising that the oncogenic potential of HSV has been examined [146]. Its potential as relates to cervical cancer has been driven by studies that detected HSV-2 RNA in cytologically abnormal cervical biopsies [147] and a DNA fragment [148] in a cervical cancer specimen. Furthermore, HSV-2 transforms cells infected with HPV-16 and HPV-18 [149]. The carcinogenic potential of both HSV-1 and HSV-2 is further suggested by studies showing the transformation of both human and mouse cells following abolishment of viral lytic capacity via photodynamic inactivation with visible light and methylene [150]. Furthermore, herpesviruses such as human cytomegalovirus have been found to adversely influence surrounding cells, for example, by inhibiting innate immunity and lengthening mitotic arrest [151]. There has been additional research examining potential mechanisms of HSV-mediated carcinogenesis [142].

G. R. B. Skinner first proposed his ‘hit and run’ mechanism of HSV-induced oncogenesis in 1976. In this experiment, primary hamster embryo fibroblasts were transformed by HSV-2 via either heat at 40 degrees Celsius or via UV-inactivated virus at 37 degrees Celsius. Cells transformed in this manner were noted to be significantly larger than the control baby hamster kidney (BHK) 21 cells. Transformed cells frequently contained dicentric chromosomes. The oncogenicity of these transformed cells was significantly higher than control cells based on tumor formation. HSV DNA was not detected in the host cell, nor was there an expression of virus-specific antigen, leading to the hypothesis that HSV-2 is a mutagen that begins the transformative process but is not involved in later steps [152]. Multiple mechanisms by which HPV may contribute to this process have been proposed, such as acting as a cofactor for HPV infection, influencing the HPV life cycle, and affecting gene expression. Such alterations would facilitate subsequent carcinogenic events mediated by HPV [153].

A second theory that attempts to explain HSV’s role in cancer development is the ‘hijacking’ hypothesis, in which viral products result in signaling pathway activation and subsequent proliferation [154]. One such viral product is the large subunit of the HSV-2 ribonucleotide reductase ICP10, which stimulates RAS activation [155]. JHLa1 cells, which constitutively express ICP10, are capable of anchorage-independent growth. These cells also have higher levels of activated ras (ras-GTP) than JHL15 cells, which express a mutant form of the ribonucleotide reductase lacking the transmembrane domain [155]. Indeed, ras is considered a proto-oncogene that is often mutated in cancer [156]. Furthermore, the ICP10 protein kinase inhibits neuronal apoptosis by activating the MEK/MAPK pathway [157,158]. Further studies demonstrate that it upregulates Bag-1, an anti-apoptotic protein [158].

### Relationship between HSV and Cervical Cancer

Though HSV is not as strongly linked to cervical cancer as HPV, several studies have aimed to explore the relationship between HSV and cervical cancer. These have yielded mixed results.

One study supporting the role of HSV in the development of cervical cancer was performed by McDougall et al. in the 1980s [159]. They discovered the presence of HSV-2 RNA in cells undergoing pre-malignant changes; however, there was no evidence of HSV-2 RNA in squamous cell cancer cells [159]. Another study performed by Li et al. examined National Health and Nutrition Examination Survey (NHANES) data from 1999 to 2014 [160]. This study found that out of HSV-1 and HSV-2, only HSV-2 was associated with the occurrence of cervical cancer [160]. It also examined the relative risk for cervical cancer and found that coinfection with both HPV and HSV-2 (relative risk [RR]_adjusted_ = 3.44) had a higher relative risk than with either HSV-2 (RR_adjusted_ = 2.79) or HPV (RR_adjusted_ = 2.98) alone [160]. Notably, a recent study of Mexican women by Bahena-Roman et al. revealed that patients seropositive for HSV-2 had a 1.7 times higher risk of having high-risk HPV than HSV-2 negative patients, and patients with high-risk HPV had a nine times higher rate of active HSV-2 infection [161]. These findings are corroborated by another recent paper demonstrating that HSV-positive patients had a much higher rate of coinfection with HPV than HSV-negative patients (adjusted odds ratio 11.032) [162]. Another recent study focusing on high-risk strains in Mali demonstrated this as well, with 56% of women with high-risk HPV strains also having HSV-2 infection compared to 37% infected with non-high-risk HPV strains [163]. The study by Bahena-Roman found an association between HSV-2 seroprevalence and active infection with cervical cancer. Indeed, the risk of having atypical squamous cells of uncertain significance (ASC-US) was 10 times greater in patients with detectable HPV DNA and high-risk HPV/HSV-2 coinfection; however, there was no increase in the risk of developing higher-risk squamous intraepithelial lesions [161]. It was concluded that HSV-2 may be involved in early stages of cell transformation but is not required for future progression to squamous cell carcinoma and that HSV-2 was not consistently noted in cervical cancer lesions [161]. This is consistent with the ‘hit and run’ hypothesis proposed by G. R. B Skinner in 1976, in which HSV-2 plays a role in initial cell transformation only [152]. Another study by de Abreu et al. found an increased risk of developing ≥ ASC-US cytology but not other cervical abnormalities such as HSIL. The authors concluded that HSV-2 may be involved in initial cellular transformation but not progression [164].

Further evidence in favor of HSV’s transformative role in cervical cancer came when DiPaolo et al. demonstrated that the *Bg*/II N region of HSV-2 was capable of converting immortalized genital epithelial cells into tumorigenic squamous cells. HK16D-1 cells were immortalized with HPV-16. Transfection with pGR62 (*Bg*/II N) transformed these cells into a tumorigenic strain, HK16D-1/HSV-2-1, which formed transplantable tumors in immunocompromised mice. The *Bg*/II N region was subsequently lost from the tumorigenic cells, although the cells’ oncogenicity was not altered [15].

There is evidence that HSV-2 infection results in increased HPV gene expression [165]. Working with HPV-16-infected CaSki cells, Pisani et al. analyzed the expression levels of E1, E2, E6, and L1 20 h post HSV-2 infection. While no significant change was noted in L1 expression, E1, E2, and E6 expression tripled when compared to mock-infected cells. There was no difference in the quantity of HPV-16 DNA [165]. In addition, HPV/HSV-2 coinfected cells overexpressed the cellular protein survivin [166]. A separate study showed that infection with either HSV-1 or HSV-2 resulted in a nearly three-fold increase in HPV-18 DNA integration into HeLa cell genomes. There was also evidence in support of HSV-1 DNA polymerase facilitating HPV-18 DNA replication [167].

One paper assessed the frequency of inflammation and cervical abnormalities in HSV-2 positive and negative patients. In total, 56.1% of HSV-2+ women in the study had inflammation, and 11.5% had cervical abnormalities when compared to 49% and 7.8%, respectively, in HSV-2 negative patients. It was concluded that HSV-2 may induce cervical inflammation, which could act as a co-factor in cervical cancer formation [168].

Furthermore, it is entirely possible that patient factors may influence the relationship between HSV and cervical cancer in some patients. Such confounding variables could explain the diverse results obtained in studies examining the relationship between HSV-2 and cervical cancer. For example, immunocompromised people may be susceptible to coinfection as well as the development of precancerous and cancerous lesions. Indeed, a recent analysis by Okoye et al. demonstrated that HSV-2 is associated with cervical lesions in women with HIV. HSV-2 (75.2% vs. 45.7%), HPV (41.9% vs. 26.7%), and squamous intraepithelial lesions (32.4% vs. 13.3%) have a higher incidence in HIV-positive individuals than HIV-negative individuals [169]. Furthermore, these patients were more susceptible to developing squamous intraepithelial lesions in the presence of viral infections than women without HIV [169]. The study by Okoye et al. indicates that not only are oncogenic viral infections, including HSV-2 and HPV, possibly associated with the development of precancerous cervical lesions, but patient factors such as immunocompromised status may facilitate the development of these lesions as well. Taku et al. recently showed similar results, demonstrating that HPV is independently associated with a higher prevalence of both HSV-2 (odds ratio 4.17) and HIV (odds ratio 2.11) [170]. Importantly, a recent study by McClymont et al. found no association between HSV-2 positivity and the incidence or persistence of HPV in women with HIV who received the HPV vaccine. There was also no association between HSV-2 positivity and precancerous lesions in this immunocompromised, vaccinated cohort. The prevalence of HSV-2 positivity remained common in these patients [171]. The study would also seem to indicate that the HPV vaccine retains efficacy in women with HIV.

Furthermore, high-risk sexual activity is associated with an increased risk of acquiring STIs [172,173]. Thus, it is also possible that patients that engage in higher-risk sexual activity have a higher risk of acquiring both HSV and HPV, which would, in turn, inflate the apparent association between the two viruses. Other studies have raised questions about the role that HSV plays in the development of cervical cancer. A meta-analysis performed by Cao et al. examined 20 studies (14 case–controls and 6 longitudinal) comprising 3337 cervical cancer patients. Cao et al. concluded that the combined odds ratio (OR) of HSV-2 patients developing cervical cancer was 1.37 (95% CI 1.12–1.69) for traditional case–control studies. The OR for prospective/retrospective nested case–control studies was 1.04 (95% CI 0.82–1.31) [174]. The authors concluded that current evidence does not support that HSV-2 plays a role in cervical cancer development [174]. Another study examining women in Nordic countries conducted by Lehtinen et al. also found that HSV-2 did not contribute to cervical cancer pathogenesis [175], and a separate study assessing Iranian women found only one instance of HSV-2 infection out of 45 patients with cervical cancer [14]. These results were replicated in a separate study of Iranian women that failed to detect HSV genomes in any of 102 patients diagnosed with low-grade squamous intraepithelial lesions (LSIL), HSIL, squamous cell carcinoma, or adenocarcinoma of the cervix [176].

Table 1 contains a summary of evidence examining the role of HSV in cervical cancer pathogenesis.

## 6. Targeting HPV and HSV for the Prevention of Cervical Cancer

Cervical cancer is one disease for which there are robust prophylactic measures in place. The mainstay of prophylaxis is the highly effective HPV vaccine [177]. Furthermore, screening can be used to identify and treat precancerous lesions, thus preventing progression to carcinoma [178].

While the relationship between cervical cancer, HPV, and HSV is not yet precisely defined, a causative relationship would open another avenue for cervical cancer prophylaxis. Both prophylactic and therapeutic treatment of HSV would reduce the incidence of genital lesions, thereby limiting HPV’s ability to access the basal layer. As was mentioned above, this access is a key step in establishing HPV infection [37]. Prophylaxis would additionally prevent any pre-malignant changes induced by HSV-2 infection.

### 6.1. Prophylaxis and Early Detection of Oncogenic HPV

As previously mentioned, vaccination against HPV is highly effective in preventing cervical cancer [177]. An analysis of 1,672,983 Swedish girls and women aged 10–30 demonstrated that cervical cancer had an incidence of 47 cases per 100,000 patients in individuals vaccinated with the quadrivalent HPV vaccine. Non-vaccinated individuals had a much higher incidence of 94 cases per 100,000 people [177]. A separate study found that the quadrivalent HPV vaccine had an efficacy of 95.4% against persistent infection with the common high-risk strains, HPV-16 and HPV-18, following a single dose [179]. A meta-analysis further confirmed the efficacy of the HPV vaccine, both in those with no prior exposure and in those previously exposed to HPV-16/18 [180]. However, there are several barriers to vaccination, including cost, limitations imposed by infrastructure, and social stigma [181]. The barriers to vaccination are particularly high in lower-income countries, given their relatively limited screening and vaccination capacities [181].

There have been three vaccines approved for use in the United States. Gardasil was approved in 2006 and provides coverage against the low-risk strains HPV-6 and HPV-11, as well as the high-risk strains HPV-16 and HPV-18 [182]. It was originally approved for females aged 9–26 and has since been approved for both genders through age 45 [182]. In 2009, the bivalent vaccine Cervarix, which protects against HPV-16 and HPV-18, was approved for females aged 10–25 [182]. In 2014, the 9-valent Gardasil 9TM (9vHPV) vaccine was approved, with protection against HPV-6, -11, -16, -18, -31, -33, -45, -52, and -58 [182]. Notably, the 9-valent vaccine has been the only vaccine used in the United States since 2016 [181]. All HPV vaccines are created from the L1 capsid protein [183], which is capable of self-assembling into immunogenic virus-like particles [183,184].

The other key component of cervical cancer prophylaxis is screening. The US Preventive Services Task Force currently recommends cytologic screening for cervical cancer every 3 years in females aged 21–29 [185]. Between the ages of 30 and 65, women can either continue cytologic screening every 3 years or undergo testing for high-risk HPV +/− cytology every 5 years [185]. Patients younger than 21 should not be screened, and there is no indication to screen patients over 65 with a sufficient history of prior screenings unless they have an increased risk of developing cervical cancer [185]. However, like with vaccines, there are barriers to the implementation of a routine screening program in both low- and high-income countries [186,187].

### 6.2. Prophylaxis and Treatment of HSV Infection

The primary therapeutic option for HSV infection is acyclovir/valacyclovir [24,25]. Many alternatives exist, including valacyclovir, penciclovir, and famciclovir. These drugs are nucleic acid analogs that interfere with viral DNA polymerase [188].

In addition to the established antivirals, there are exciting new therapies emerging as potential treatment options. One such compound is ginkgolic acid, a product of the Ginkgo biloba tree. This compound has been shown to inhibit fusion across multiple classes of viral fusion [189]. HSV-1 was among the viruses inhibited by ginkgolic acid [189], and it is not unreasonable to assume that HSV-2 may be inhibited in a similar fashion. Brincidofovir, a lipid conjugate of cidofovir, was shown to act synergistically with acyclovir to inhibit HSV replication both in culture and murine models [190]. More recently, its efficacy in preventing breakthrough HSV infection was assessed in the setting of hematopoietic stem cell transplant recipients. Initial results were promising, with a reported rate of 1.0 per 1000 patient days [191]. Amenamevir is a helicase-primase inhibitor that showed comparable efficacy to valacyclovir in treating mice cutaneously inoculated with HSV-1. Furthermore, it demonstrated efficacy when applied on day 4 post-infection, something not seen in mice treated with valacyclovir [192].

Prophylactic measures against HSV are also under research. For example, Dropulic et al. administered the replication-deficient HSV-2 vaccine HSV529 to adults without either HSV-1 or HSV-2, with HSV-2 and with or without HSV-1, and with HSV-1 but without HSV-2. In total, 78% of patients without either virus had at least a four-fold increase in their titers of neutralizing antibodies, a response not seen in either of the other groups. In previously seronegative patients, CD4+ T cell responses were noted in 36% of patients compared to 46% in the group infected by HSV-2 +/− HSV-1 infection and 27% of the patients infected by HSV-1 but not HSV-2. CD8+ T cell responses were noted in 14%, 8%, and 18% of patients in these groups, respectively [193]. mRNA vaccines have also been assessed as prophylactic measures against HSV-2 in murine models. Mice vaccinated with an mRNA vaccine, including glycoproteins gC, gD, and gE, did not develop genital disease following challenge with HSV-1 or HSV-2. Vaginal swabs were positive for HSV-1 in 4/10 and for HSV-2 in 0/5 mice at the lowest challenge dose 5 × 10^4^ plaque-forming units. Day 2 swabs were positive for HSV-1 in 5/5 mice and for HSV-2 in 3/5 mice at 2 × 10^5^ plaque-forming units. Swabs were positive for HSV-1 in 12/15 and for HSV-2 in 5/10 mice at 2 × 10^6^ plaque-forming units. Only 3/30 HSV-1-infected mice and 1/20 HSV-2-infected mice had positive titers 4 days after infection. Importantly, no HSV DNA was detected in the dorsal root ganglia of any mouse that received the mRNA vaccine [194].

The VOICE trial is a clinical trial that assessed the efficacy of tenofovir, an adenine nucleotide analog reverse transcriptase inhibitor, in preventing the acquisition of HSV-2. Patients were instructed to apply tenofovir 1% gel once daily. There was a trend towards a reduced risk of seroconversion with HSV-2, although the results were not statistically significant, with a hazard ratio of 0.60 (95% CI, 0.33–1.08; *p* = 0.086). This ratio was adjusted for location, age, HIV status, hormonal contraception use, having at least two male sex partners in the last 3 months, and having anal sex in the past 3 months [195].

## 7. Recent Advances in Cervical Cancer Therapeutics

In spite of these preventative and screening measures, cervical cancer still impacted 604,000 women globally in the year 2020 [1]. Treatment options are dependent on the stage of disease and include surgery, radiation therapy, including external beam therapy and/or brachytherapy, and chemotherapy/immunotherapy [196]. Most early cervical cancers can be successfully treated with surgical intervention [196].

### 7.1. Systemic Therapies

Many recent advances in cervical cancer therapy have been in the field of immunotherapy [197]. Programmed Cell Death Protein 1 (PD-1) expression leads to the inhibition of the immune system by downregulating the T cell response via promoting the apoptosis of antigen-specific T cells and the inhibition of regulatory T cell (Treg) apoptosis [198]. Programmed Cell Death Ligand 1 (PD-L1) is another immune inhibitory molecule that is frequently implicated in dampening the host immune response to malignancies [198]. PD-L1 is expressed in a subset of cervical cancers. In fact, a recent experiment demonstrated that 56% of cases expressed PD-L1 mRNA while 41% of cases expressed the PD-L1 protein [199]. Pembrolizumab is a humanized monoclonal anti-PD-1 antibody that has shown promise in a variety of malignancies [200] and has FDA approval for the treatment of advanced PD-L1 cervical cancer [201]. This drug is associated with relatively few serious adverse events [200].

The efficacy of pembrolizumab in patients with previously treated advanced cervical cancer was recently assessed with patients enrolled in the phase II KEYNOTE-158 study [202]. The overall response rate (ORR) in a 98-patient sample was 12.2%, including three complete responses. Of the 12 responding patients, 11 had stage IVB disease and 1 had stage IIIB disease. Nine patients had durable responses of at least 9 months. The ORR in patients with PD-L1+ tumors was 14.6% [202]. A separate study of patients with persistent, recurrent, or metastatic cervical cancer showed that median progression-free survival in patients with PD-L1+ tumors was 10.4 months in the pembrolizumab arm and 8.2 months in the placebo arm. Overall survival was also significantly longer in the pembrolizumab arm, with a 24-month estimate of 53.0% in patients taking pembrolizumab versus 41.7% in patients receiving placebo. Both groups received platinum-based chemotherapy +/− bevacizumab [203].

Nivolumab is a fully human (monoclonal antibody) PD-1 inhibitor used in a variety of cancers [204]. In total, 19 patients with metastatic or recurrent cervical cancer in the phase I/II CheckMate 358 trial received nivolumab monotherapy. All had been previously treated. The objective response rate was 26.3%, and the median response duration had not been attained by 19.2 months [205]. A separate study of twenty-five patients showed a partial response in one patient with stable disease in an additional nine patients. Progression-free survival and overall survival were estimated to be 16% and 78.4%, respectively, after 6 months [206].

Vascular endothelial growth factor (VEGF) is a major contributor to angiogenesis [207]. A recent trial assessed the efficacy of the tyrosine kinase inhibitor of VEGF receptor 2 (VEGFR2), apatinib, in conjunction with camrelizumab, a fully humanized PD-1 antibody, in the setting of advanced cervical cancer. In a cohort of 45 patients, the ORR was 55.6%, including two patients who experienced a complete response and 23 patients who experienced a partial response. The median progression-free survival was 8.8 months [208].

CTLA-4 is another T cell inhibitory molecule and is expressed by Tregs; thus, CTLA-4 blockade represents a means by which to enhance the host immune response to tumors [209]. Ipilimumab is a fully human [210] monoclonal antibody that acts as a CTLA-4 inhibitor [211]. Mayadev et al. analyzed the efficacy of ipilimumab following chemoradiation in 21 patients with node-positive cervical cancer. Patients first received concurrent radiotherapy and cisplatin, followed by ipilimumab. The overall 12-month survival was 90%, and progression-free survival was 81% [212]. The efficacy of ipilimumab in combination with nivolumab has also been assessed in patients with recurrent/metastatic cervical cancer [213]. Patients received either nivolumab 3 mg/kg every 2 weeks + ipilimumab 1 mg/kg every 6 weeks (A) or four doses of nivolumab 1 mg/kg + ipilimumab 3 mg/kg every 3 weeks followed by nivolumab 240 mg every 2 weeks (B) for a total of 24 months. ORR was higher in B. Progression-free survival was 13.8 months in A and 8.5 months in B in treatment-naïve patients. Progression-free survival was 3.6 months in A and 5.8 months in B in patients who had received prior systemic therapy. Overall survival was not reached in either A or B in patients without prior treatment and was 10.3 months in A and 25.4 months in B for patients who had received prior therapy [213].

Tissue factor is a key component of the clotting cascade that is widely expressed in malignancy [214]. In cancer, tissue factor promotes hematogenous metastasis, angiogenesis, and tumor growth, among other things [214]. Tisotumab vedotin, a fully human antibody with a tolerable safety profile [215], is an antibody-drug conjugate targeting tissue factor [216] whose efficacy was assessed in patients with recurrent or metastatic cervical cancer [216]. The ORR in 101 patients was 24%, including 7 complete responses and 17 partial responses. In total, 28% of patients experienced at least grade 3 side effects [216].

An advance in systemic therapy for advanced cervical cancer occurred in 2017 with the Gynecology Oncology Group 240 trial [217] assessing the humanized antibody [218] bevacizumab, an angiogenesis inhibitor, in metastatic, persistent, or recurrent cervical cancer [217]. Other chemotherapeutic agents used in this trial included cisplatin, which induces DNA damage [219], paclitaxel, which promotes microtubule assembly and inhibits microtubule disassembly [220], and topotecan, which inhibits topoisomerase I [221]. Patients were treated in one of four arms: cisplatin + paclitaxel, topotecan + paclitaxel, cisplatin + paclitaxel + bevacizumab, and topotecan + paclitaxel + bevacizumab. Groups that received bevacizumab had increased survival relative to those that did not (16.8 months in patients receiving bevacizumab v. 13.3 months in patients not receiving bevacizumab) [217].

### 7.2. Other Therapeutic Advances

Per the National Comprehensive Cancer Network (NCCN) guidelines, the current standard of care for radiation therapy includes coverage of any gross disease (including at least a 3 cm margin from any identifiable disease), parametria, uterosacral ligaments, and high-risk nodal areas, including at least the external iliac, internal iliac, obturator, and presacral nodal basins. More extensive coverage may be necessary based on clinical suspicion. Larger fields are also recommended in the case of disease extension to the iliac/para-aortic nodes or the lower 1/3 of the vagina. Microscopic disease requires doses of 40–45 Gy with boosts to areas of gross disease, including a brachytherapy boost to the primary tumor in patients who are not operative candidates [222].

There have been recent advances in the use of radiation therapy to treat cervical cancer. Patients with FIGO stage IB-IVA disease treated with curative intent were given chemoradiotherapy with cisplatin. They were then treated with image-guided adaptive brachytherapy (IGABT) using MRI treatment localization [223]. The 5-year actuarial local control rate was 92% (median follow-up of 51 months). The authors concluded that chemoradiotherapy followed by IGABT resulted in stable control of local disease in locally advanced cervical cancer [223].

The efficacy of brachytherapy was also assessed in the setting of pelvic recurrence. Patients received either brachytherapy or radiotherapy to the entire pelvis with concurrent cisplatin therapy [224]. The effective control rate was 76.7%, 80.0%, 83.3%, and 86.7% in the brachytherapy group at 1,3,6, and 12 months with an overall median survival of 4.34 years. The chemoradiotherapy group had effective control rates of 65.6%, 65.5%, 62.5%, and 71.9% at the same time points, with an overall median survival of 3.59 years [224].

A clinical trial performed by Huang et al. assessed the benefit of sequential chemoradiation (SCRT) and concurrent chemoradiation (CCRT) versus radiation alone in the post-operative setting [225]. Patients with FIGO stage IB to IIA cervical cancer received radiation alone, CCRT, or SCRT. In a cohort of 1048 patients, 172 patients had either recurrent disease or died of cervical cancer, including 19.4% of the 350 patients in the radiation alone group, 12.5% of the 353 patients in the SCRT group, and 17.4% of the 345 patients in the CCRT group. SCRT had significantly better disease-free survival than CCRT and radiation alone. The SCRT group also had a lower rate of distant metastasis than the other groups [225]. This contrasts with the results of the OUTBACK trial, a phase III trial assessing the efficacy of chemoradiotherapy containing cisplatin compared to chemoradiotherapy followed by four cycles of adjuvant carboplatin and paclitaxel. The 5-year survival rate was almost identical between groups (72% in the group receiving adjuvant chemotherapy and 71% in the group receiving chemoradiotherapy alone). The adjuvant chemotherapy group had a higher rate of treatment-related complications, particularly infection [226].

In an interesting development, viruses such as HSV are being harnessed to treat cervical cancer. An interesting link between HSV and cervical cancer comes in the form of oncolytic herpes simplex viral therapy (OHSVT). Oncolytic viruses can target abnormal cells through tumor-associated targets such as a prostate-specific antigen, human telomerase reverse transcriptase, HER2/neu, and the endothelial growth factor receptor, among others. The virus then promotes the destruction of tumor cells by activating the immune system against the infected tumor cells and through direct cytopathic damage [227]. T-01 is an oncolytic herpes simplex virus whose efficacy was recently assessed in the context of HPV-associated cervical cancer [228]. In vitro experiments showed that T-01 exerted both cytotoxic and cytopathic effects on the human cervical cancer lines HeLa, CaSki, and SKG-IIIa within 24 h post-infection [228]. T-01 was then tested in NOD-SCID mice following xenograft implantation of human HeLa cells. The tumor was eliminated in four of six mice examined, and the tumor regression rate was 83.3%. Similar marked improvements were noted in C57BL/6 mice implanted with the murine cervical cancer line TC-1 [228]. TC-1-implanted mice treated with T-01 had higher levels of CD8+ T cells than the control [228].

Other oncolytic viruses are being assessed in the setting of cervical cancer. Ni et al. compared the efficacy of the oncolytic vaccinia virus in the presence and absence of *aphrocallistes vastus* lectin (AVL), a C-type lectin produced by sponges [229]. Oncolytic vaccinia virus with AVL (oncoVV-AVL) had greater cytotoxic effects on HeLa and HeLa S3 cells than vaccinia virus without AVL. OncoVV-AVL was shown to inhibit tumor growth and augment vaccinia replication in HeLa S3 cells. It also demonstrated a more potent antitumor effect in xenograft Balb/c mice that had HeLa S3 cells subcutaneously injected [229]. Sato et al. used an adeno-associated virus vector (AAV) to deliver short hairpin RNA (shRNA) targeting E6 and E7 (termed AAV2-shE6E7). In in vitro studies, transfection of AAV2-shE6E7 resulted in increased apoptosis, and in vivo mouse studies demonstrated that a single dose of AAV2-shE6E7 resulted in significantly decreased tumor volume compared to controls [230].

E6 and E7 have also emerged as potential targets for oncolytic vaccines. As the main contributors to HPV-induced carcinogenesis, these proteins represent logical targets. Multiple studies have examined the efficacy of these vaccines ([81]), including a phase 2b trial assessing the therapeutic synthetic vaccine VGX-3100, which targets the E6 and E7 proteins of HPV 16 and HPV18 in cervical intraepithelial neoplasia (CIN) 2 and 3. In this trial, nearly 50% of the recipients demonstrated histopathologic regression at 36 weeks compared to approximately 30% for the placebo group [231]. In a longer-term analysis, 91% of patients whose HSIL regressed without surgical intervention had no evidence of HPV16 or 18 at 18 months. Patients who received a placebo followed by surgery had similar rates of resolution, indicating that VGX-3100 could be a viable alternative to surgery for patients with CIN 2 and CIN 3 [232]. Vvax001 is a vaccine that uses replication-incompetent Semliki Forest virus replicon particles that encode the E6 and E7 antigens from HPV16. In a phase I trial, this vaccine was shown to induce CD4 and CD8 responses against E6 and E7. The vaccine was well tolerated, with the only adverse effect being mild reactions at the injection site [233]. Youn et al. examined the efficacy of GX-188E, which encodes the E6 and E7 proteins from HPV16 and HPV18, in combination with pembrolizumab in patients with advanced HPV16+ or HPV18+ cervical cancer. In total, 26 patients were evaluated. After 24 weeks, eleven patients experienced a response, including four complete responses and seven partial responses [234].

A more comprehensive review of updates in the management of cervical cancer can be found elsewhere [196]. Table 2 summarizes recent advances in cervical cancer therapy.

## 8. Conclusions

Despite its preventable nature, cervical cancer continues to represent a significant cause of morbidity and mortality worldwide, particularly in countries that lack the means for extensive vaccination and screening programs. In this review, we discussed the role of HPV in the development of cervical cancer and explored the evidence for and against HSV involvement in the pathogenesis of cervical cancer. We also provided an overview of these two viruses and discussed recent therapeutic advances in cervical cancer. While the link between HPV and cervical cancer is definitive, there is a paucity of research analyzing the potential role of HSV in cervical cancer. Evidence that does support a role for HSV in the pathogenesis of cervical cancer seems to indicate that the most likely mechanism is the ‘hit and run’ hypothesis, in which HSV infection is involved in initiating the oncogenic process but is not involved in later steps. However, a definitive link has yet to be established, and other studies have cast doubt on the involvement of HSV in cervical cancer pathogenesis. The association between HSV and cervical cancer, specifically HSV-2, has not yet been unambiguously established. Even if an association was ultimately established, it would be necessary to demonstrate the causative role of HSV-2, as such an association could potentially be attributed to shared immunological compromise, genetic predisposition, or lifestyle factors. Therefore, more research is required to further elucidate the potential relationship between HSV and cervical cancer, and the extent of HSV involvement, if any, in order to establish a definitive link between the two.

## Figures and Tables

**Figure 1 cancers-15-03692-f001:**
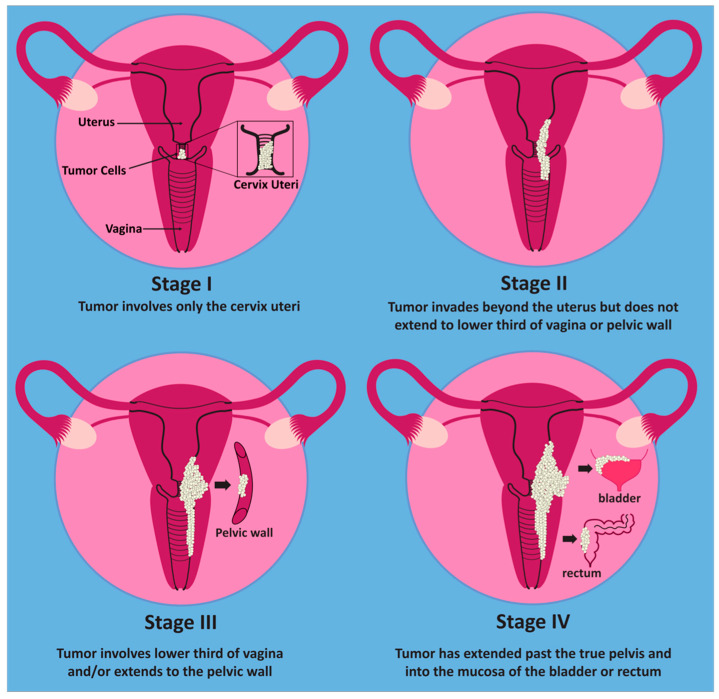
Stages of cervical cancer. In stage I, the tumor only involves the cervix. In stage II, the tumor extends beyond the cervix but does not yet involve the lower third of the vagina or pelvic wall. Stage III involves the lower third of the vagina and/or the pelvic wall. Once the tumor has spread beyond the true pelvis or involves the mucosa of the bladder or rectum, it is considered stage IV.

**Figure 2 cancers-15-03692-f002:**
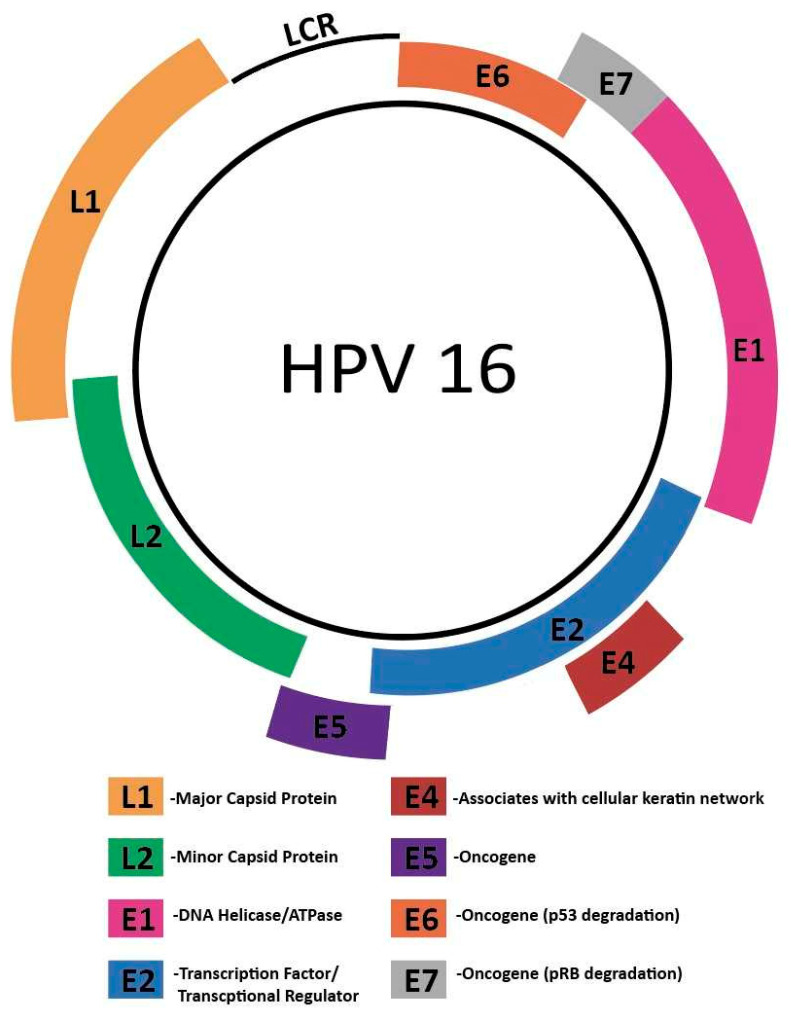
Genome structure of HPV16. The HPV genome includes a long control region (LCR), an early gene region encoding E1, E2, E4, E5, E6, and E7, and a late gene region encoding L1 and L2.

**Figure 3 cancers-15-03692-f003:**
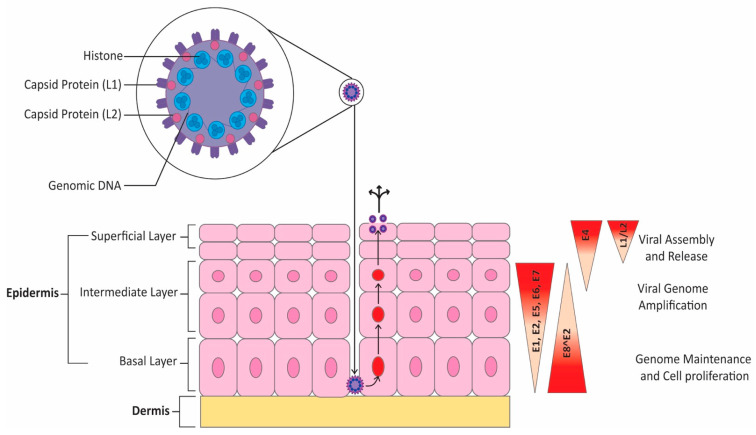
HPV life cycle. HPV infection begins after disruptions in the epithelial barrier allow viral access to the basal layer. Viral gene expression is tightly linked with the level of epithelial cell differentiation. Early in the infection, only the early promoter is active. HPV episomes are then maintained in undifferentiated, dividing cells. Upon differentiation, HPV enters the productive phase of the viral cycle, which results in expression of late genes. The process culminates in terminally differentiated cells, which includes expression of L1 and L2 as well as viral assembly and release.

**Figure 4 cancers-15-03692-f004:**
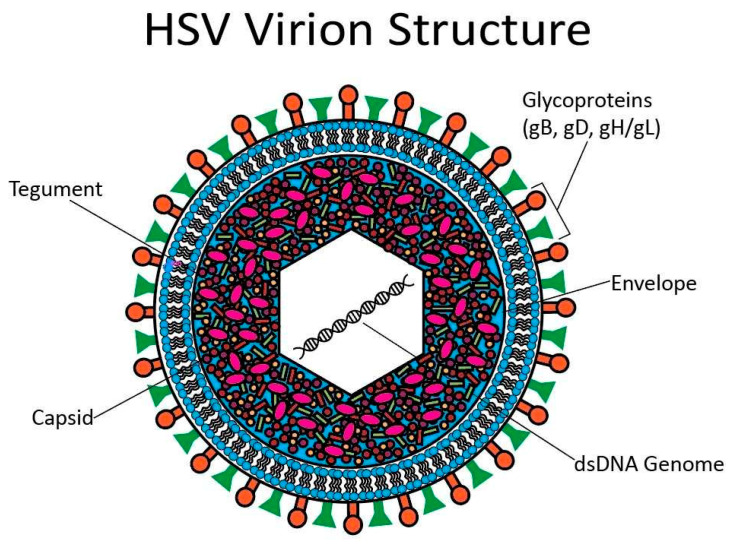
HSV virion structure. At the center of the HSV virion is a linear, dsDNA genome surrounded by an icosahedral capsid. There are tegument proteins around the capsid. A lipid bilayer studded with glycoproteins forms the outermost layer.

**Table 1 cancers-15-03692-t001:** Summary of proposed relationships between HSV, HPV, and cervical cancer.

Study	Relationship Exists	Relationship Uncertain	Findings	Reference
McDougall et al.		X	Presence of HSV-2 RNA in cells undergoing pre-malignant changes; no evidence of HSV-2 RNA in squamous cell cancer cells	[159]
Li et al.	X		HSV-2 associated with the occurrence of cervical cancer; coinfection with both HPV and HSV-2 had a higher relative risk than either infection with HSV-2 or HPV alone	[160]
Bahena-Román et al.	X		Patients seropositive for HSV-2 had a higher risk of high-risk HPV; patients with high-risk HPV had a higher rate of active HSV-2 infection	[161]
de Abreu et al.	X		Increased risk of developing ≥ ASC-US cytology (initial cellular transformation) but not other cervical abnormalities such as HSIL. The authors concluded that HSV-2 may be involved in initial cellular transformation but not progression	[164]
DiPaolo et al.	X		Bg/II N region of HSV-2 converted immortalized genital epithelial cells into tumorigenic squamous cells	[15]
Pisani et al.	X		HSV-2 infection increased HPV gene expression of E1, E2 and E6 in HPV-16-infected cells	[165]
Paba et al.	X		HPV/HSV-2 coinfected cells overexpressed cellular protein survivin	[166]
Hara et al.	X		HSV-1 DNA polymerase could facilitate HPV-18 DNA replication	[167]
Koanga et al.	X		HSV-2 infection may induce cervical inflammation, potentially acting as a co-factor in cervical cancer formation	[168]
Okoye et al.	X		Immunocompromised status may facilitate viral-mediated oncogenesis	[169]
Cao et al.		X	Current evidence does not support the idea that HSV-2 plays a role in cervical cancer development	[174]
Lehtinen et al.		X	HSV-2 did not contribute to cervical cancer pathogenesis in Nordic countries	[175]
Ahmadi et al.		X	Only one instance of HSV-2 infection found out of 45 patients with cervical cancer	[14]
Joharinia et al.		X	No detection of HSV genomes in any patients diagnosed with low-grade squamous intraepithelial lesions, high-grade squamous intraepithelial lesions, squamous cell carcinoma or adenocarcinoma of the cervix	[176]

**Table 2 cancers-15-03692-t002:** Updates On Cervical Cancer Therapy.

Drug	Mechanism of Action	References
Pembrolizumab	PD-1 inhibitor	[200,201,202,203]
Nivolumab	PD-1 inhibitor	[204,205,206,213]
Apatinib + camrelizumab	Apatinib: Tyrosine kinase inhibitor of VEGFR2 Camrelizumab: PD-1 inhibitor	[208]
Ipilimumab	CTLA-4 inhibitor	[211,212,213]
Nivolumab + ipilimumab	Nivolumab: PD-1 inhibitor Ipilimumab: CTLA-4 inhibitor	[213]
Tisotumab vedotin	Antibody/drug conjugate targeting tissue factor	[215,216]
Bevacizumab	Angiogenesis inhibitor	[217,218]
Brachytherapy	Radiation	[223,224]
SCRT v. CCRT v. RT	N/a	[225]
Oncolytic viruses	Cytopathic and cytotoxic killing of tumor cells	[227,228,229,230]
Vaccine therapies	Targeting E6/E7	[81,231,232,233,234]

## Data Availability

All papers cited in this review can be found on either PubMed or Google Scholar.

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
