# Peer review of "Herpes Simplex Virus, Human Papillomavirus, and Cervical Cancer: Overview, Relationship, and Treatment Implications"

_cancers, 2023, doi:10.3390/cancers15143692_

Round 1

Reviewer 1 Report

HSV, HPV, and Cervical Cancer: What is the Relationship?

Overall this is a well written review. I am not sure of the relationship between the title / stated aims in the abstract and the inclusion of parts 6 and 7 - I think these should be removed or the aim / title adjusted/reworked.

Abstract

1.     No comments

Introduction

1.     Figure one is not very clearly for cervical cancer – Stage 1 appears to be in the endometrium rather than the cervix, stage 2 the image of the cancer does not extend beyond the uterus, stage 3 the image involves neither the lower 1/3 of the vagina nor the pelvic side wall

Overview of HSV infection

1.     “HSV-2are” needs a space before are

Relationship between HSV and cervical cancer

1.     The authors should address the possibility that the relationship is in the patient factors (e.g. more immunosuppression increasing likelihood of both HSV and HPV) rather than in the HSV increasing HPV as a causality step and that this may explain the heterogenous results of the included studies

Given the title of the paper I am unclear on the reason for including parts 6 and 7

The EMBRACE trial should be prefaced with the current gold standard treatment approaches

If the Huang trial is going to be included then the OUTBACK trial needs to be set alongside this

The discussion at the beginning of page 16 about the theory for the link between HSV and cervical cancer seems out of place and has no relationship with the treatment paragraph that precedes it / the heading it is included under

Author Response

We would like to thank the reviewer for the insightful comments. Our responses are in-line as follows:

Overall this is a well written review. I am not sure of the relationship between the title / stated aims in the abstract and the inclusion of parts 6 and 7 - I think these should be removed or the aim / title adjusted/reworked.

Line 560-601: A section on prevention, prophylaxis, and treatment of HSV infection and reducing HPV infection was included.

The title was changed to “HSV, HPV, and Cervical Cancer: 

Overview, Relationship, and Treatment Implications” and the abstract was altered to better represent the body of the paper. 

Introduction

  1. Figure one is not very clearly for cervical cancer – Stage 1 appears to be in the endometrium rather than the cervix, stage 2 the image of the cancer does not extend beyond the uterus, stage 3 the image involves neither the lower 1/3 of the vagina nor the pelvic side wall

Figure 1 was updated to more accurately reflect the invasion patterns of cervical cancer.

Overview of HSV infection

  1. “HSV-2are” needs a space before are

Line 219: A space was added between HSV-2 and ‘are’.

Relationship between HSV and cervical cancer

  1. The authors should address the possibility that the relationship is in the patient factors (e.g. more immunosuppression increasing likelihood of both HSV and HPV) rather than in the HSV increasing HPV as a causality step and that this may explain the heterogenous results of the included studies

Line 471-492: A discussion of the possibility that patient factors influence the relationship between HSV and cervical cancer was included.

Given the title of the paper I am unclear on the reason for including parts 6 and 7

In addition to information covered in previous reviews, our review gives up-to-date, relevant data and guidelines regarding prevention, early detection and treatment of oncogenic HPV, HSV infection, and cervical cancer. 

Line 528-539: A section on the importance of prophylaxis in cervical cancer was added.          

Line 560-601: A section on prevention, prophylaxis, and treatment of HSV infection and reducing HPV infection was included.

The title was changed to “HSV, HPV, and Cervical Cancer: Overview, Relationship, and Treatment Implications” and the abstract was altered to better represent the body of the paper. 

The EMBRACE trial should be prefaced with the current gold standard treatment approaches

Line 692-698: Gold standard radiotherapeutic approaches were included prior to the EMBRACE trial.

If the Huang trial is going to be included then the OUTBACK trial needs to be set alongside this

Line 714-720: The OUTBACK trial was included.

The discussion at the beginning of page 16 about the theory for the link between HSV and cervical cancer seems out of place and has no relationship with the treatment paragraph that precedes it / the heading it is included under

Line 721-723: The introduction to oncolytic viral therapy was altered so that it flows better with the preceding paragraph and subheading.

Reviewer 2 Report

The present review article titled “HSV, HPV, and Cervical Cancer: What is the Relationship?” by Sausen et al. is very well written. The authors cover almost all the aspects as per the rationale of the manuscript. The flow as well as linguistic quality is excellent. Overall, the quality of the manuscript is excellent. The author needs to address the following comment.

Comment 1. The abstract section is short and not stand-alone. The authors should provide a complete background of the manuscript.

Comment 2. The authors should improve the quality of the figures, especially Figure 2. 

Author Response

We would like to thank the reviewer for the insightful comments. Our responses are in-line as follows:

The abstract section is short and not stand-alone. The authors should provide a complete background of the manuscript.

The title and abstract were altered to better represent the body of the paper.

The authors should improve the quality of the figures, especially Figure 2. 

Higher quality figures were included in the paper. 

Round 2

Reviewer 1 Report

Previous comments addressed

Author Response

We thank the reviewer!